# Synthesis, Photoluminescence and Vibrational Properties of Aziridinium Lead Halide Perovskites

**DOI:** 10.3390/molecules27227949

**Published:** 2022-11-17

**Authors:** Dagmara Stefańska, Maciej Ptak, Mirosław Mączka

**Affiliations:** Institute of Low Temperature and Structure Research, Polish Academy of Sciences, Okólna 2, 50-422 Wrocław, Poland

**Keywords:** hybrid organic–inorganic perovskites, lead halides, aziridinium, photoluminescence, Raman

## Abstract

Three-dimensional lead halide perovskites are known for their excellent optoelectronic properties, making them suitable for photovoltaic and light-emitting applications. Here, we report for the first time the Raman spectra and photoluminescent (PL) properties of recently discovered three-dimensional aziridinium lead halide perovskites (AZPbX_3_, X = Cl, Br, I), as well as assignment of vibrational modes. We also report diffuse reflection data, which revealed an extended absorption of light of AZPbX_3_ compared to the MA and FA counterparts and are beneficial for solar cell application. We demonstrated that this behavior is correlated with the size of the organic cation, i.e., the energy band gap of the cubic lead halide perovskites decreases with the increasing size of the organic cation. All compounds show intense PL, which weakens on heating and shifts toward higher energies. This PL is red shifted compared to the FA and MA counterparts. An analysis of the PL data revealed the small exciton binding energy of AZPbX_3_ compounds (29–56 meV). Overall, the properties of AZPbX_3_ are very similar to those of the well-known MAPbX_3_ and FAPbX_3_ perovskites, indicating that the aziridinium analogues are also attractive materials for light-emitting and solar cell applications.

## 1. Introduction

Hybrid organic–inorganic compounds (HOIPs) have been extensively investigated in recent years due to their attractive functional properties [1,2,3,4]. In particular, three-dimensional (3D) lead halide perovskites of general formula APbX_3_ (A = organic cation, X = Cl^−^, Br^−^, I^−^) have recently emerged as candidates for photovoltaic, light-emitting, lasing and scintillating applications [2,4,5,6,7,8,9,10]. Their attractiveness stems from high one- and two-photon absorption coefficients, long carrier diffusion lengths and tunable band gaps [2,8,11,12]. Unfortunately, the small size of the perovskite cavity should allow accommodation of only the smallest organic cations. Moreover, many lead halides comprising small cations, such as hydroxylammonium, hydrazinium, azetidinium and imidazolium, do not crystallize in the 3D perovskite structure [13,14,15,16]. As a result, 3D structures were found only for methylammonium (CH_3_NH_3_^+^, MA^+^) [17], formamidinium (NH_2_CHNH_2_^+^, FA^+^) [17,18] and methylhydrazinium (CH_3_NH_2_NH_2_^+^, MHy^+^) cations [8,9]. It is worth adding that MHyPbX_3_ perovskites (X = Br^−^, Cl^−^) differ from their centrosymmetric MA^+^ and FA^+^ counterparts since their low-temperature (LT) phases are polar; that is, they crystallize in chiral space group *P*2_1_ [8,9,10]. Therefore, these compounds exhibit second harmonic generation (SHG) and pyroelectric properties [8,9,10].

In order to understand the optical properties of lead halide perovskites and obtain information on exciton binding energy, it is important to study their electronic absorption and PL in a broad temperature range. PL studies of MAPbX_3_ and FAPbX_3_ have been reported in many papers, and these spectra consist of narrow bands near 402–413, 532–567 and 756–830 nm for the Cl, Br and I analogues, respectively, which were attributed to free excition (FE) recombination [19,20,21,22,23,24,25]. It is worth adding that, in many cases, additional narrow bands were observed at slightly lower energies, which were attributed to defects [21,23]. A recent study of MAPbI_3_ suggested, however, that for this compound, the lower energy band near 830 nm corresponds to FE recombination, while surface defects lead to appearance of a higher energy and weaker band near 780 nm [25]. In contrast to MAPbX_3_ and FAPbX_3_ compounds, MHyPbX_3_ analogues also showed the presence of additional broad and highly Stokes-shifted bands, which could be attributed to self-trapped excitons (STEx) [8,9,10]. The presence of these bands is consistent with large distortion and off-center displacement of Pb^2+^ in these compounds [8,9,10], leading to a large increase of the electron–phonon interaction [10].

It is also important to understand phonon properties since a key phenomenon relevant for the optoelectronic applications of hybrid perovskites is electron–phonon coupling, which depends on vibrational energies and lattice dynamics [26,27,28]. Therefore, MA-, FA- and MHy-based perovskites have been the subject of numerous Raman studies [8,27,29,30,31,32,33,34,35]. In some cases, IR and THz spectra were also reported [32,34,36,37]. These studies showed that internal modes of organic cations are usually observed above 300 cm^−1^, and these bands are much weaker than the majority of lattice modes, which appear below 250 cm^−1^ [8,27,29,30,31,32,33,34,35].

It has been very recently reported that 3D perovskites can also be obtained by using aziridinium (CH_2_CH_2_NH_2_^+^, AZ^+^) cation [38]. At room temperature (RT), these compounds crystallize in the cubic symmetry (space group *Pm*$\bar{3}$*m*), and their band gaps were estimated as 2.99, 2.27 and 1.52 eV for the Cl, Br and I analogue, respectively [38]. This report also showed IR spectra, but no analysis of the obtained spectra and assignment of bands was proposed [38].

Herein, we report the synthesis of these compounds through the use of a different method than previously reported, reinvestigate electron absorption of these compounds and report temperature-dependent PL data to obtain information on exciton binding energy. We also report RT Raman spectra and propose the assignment of the observed bands to respective vibrations of atoms.

## 2. Results and Discussion

### 2.1. Raman Spectra

Vibrational modes of the 3D perovskites can be subdivided into vibrations of inorganic framework and lattice modes of organic cations (translational and librational modes) and internal modes of organic cations [8,27,29,30,31,32,33,34,35,39,40,41]. The former Raman studies of 3D perovskites showed that the vibrations of the inorganic frameworks give rise to very strong bands in the low-wavenumber region [8,27,29,30,31,32,33,34,35,39,40,41]. We assign, therefore, the very strong and asymmetric band at 139–122 cm^−1^ to Pb-X stretch (Figure 1 and Appendix A). The second low-wavenumber band is very sensitive to the type of halogen anion; that is, it shifts from 468 cm^−1^ for AZPbCl_3_ to 308 cm^−1^ for AZPbBr_3_ and 240 cm^−1^ for AZPbI_3_ (Figure 1 and Appendix A). Very similar behavior was previously reported for the MA-, FA- and MHy-based 3D perovskites, which showed the corresponding bands at 472 and 481 cm^−1^ for MAPbCl_3_ and MHyPbCl_3_, respectively; at 323, 307 and 309 cm^−1^ for MAPbBr_3_, FAPbBr_3_ and MHyPbBr_3_, respectively; and at 241 and 237 cm^−1^ for MAPbI_3_ and FAPbI_3_, respectively [34]. Studies of MAPbX_3_ perovskites showed that this mode, often assigned to the torsional mode of MA^+^ [32,39], is not a pure torsion but a mixed mode that also involves breathing or rocking motions of hydrogen atoms [42]. Another rationale why this mode cannot be attributed to a pure torsion is its very large width of the Raman band and the fact that it should be Raman-inactive [32,41]. Furthermore, this mode was shown to couple with the inorganic cage through N−H·X hydrogen bonds [34,41,42]. Due to special character of this mode, Nakada et al. proposed to call it MA-cage mode [29]. It is worth adding that theoretical calculations performed for an isolated MHy^+^ cation also showed the mixed character of the lowest wavenumber modes near 250 cm^−1^, with large contribution of the NH_2_ torsion [43]. Although no theoretical data reporting vibrational modes of AZ^+^ cation are available, the fact that the discussed modes of AZPbX_3_ compounds have similar frequencies to MA-, FA- or MHy-cage modes; they strongly depend on the type of halogen anion; and the Raman bands are very broad strongly support the assignment of the 468–240 cm^−1^ bands to the AZ-cage modes.

The remaining modes can be attributed to internal vibrations of AZ^+^ cation. Free AZ^+^ cation has 21 vibrational degrees of freedom. Six of them correspond to symmetric stretching (ν_s_NH_2_), antisymmetric stretching (ν_as_NH_2_), scissoring (δNH_2_), rocking (ρNH_2_), wagging (ωNH_2_) and torsion or twisting (τNH_2_) modes of the NH_2_ groups. The remaining 15 modes can be subdivided into symmetric stretching (2ν_s_CH_2_), antisymmetric stretching (2ν_as_CH_2_), scissoring (2δCH_2_), rocking (2ρCH_2_), wagging (2ωCH_2_) and torsion or twisting (2τCH_2_) modes of the CH_2_ groups, as well as ring stretch and two ring deformation modes.

The vibrational modes of aziridine molecule were discussed in a few papers, which reported IR and Raman spectra of vapor and liquid aziridine, as well as ab initio calculations [44,45,46]. Based on these studies, we can unambiguously assign the narrow Raman bands near 3010–3130 cm^−1^ to the νCH_2_ modes (Figure 1 and Appendix A). The Raman spectra also show a few weak and broad bands in the 3150–3230 cm^−1^ range (Appendix A). Their large width and the fact that they correspond to very broad and strong IR bands [38] indicate that they correspond to the νNH_2_ modes. It is worth noting that these bands are observed at higher wavenumbers than the corresponding bands of MAPbX_3_ perovskites (3100–3190 cm^−1^; see Reference [34]), indicating that AZ^+^ cations form weaker hydrogen bonds with the halide anions than MA^+^ cations. The δNH_2_ mode is observed as a broad band near 1540–1550 cm^−1^ (Figure 1 and Appendix A). The band near 1460 cm^−1^ with a shoulder near 1440 cm^−1^ corresponds to the δCH_2_ modes, and the most intense band near 1220–1230 cm^−1^ can be attributed to the ring stretch [46]. Bands in the 760–1220 cm^−1^ range are more difficult to assign since the assignment of modes proposed in the literature is based on studies of aziridine molecule, not aziridinium cation, and therefore the energy of some modes may be significantly modified after protonation of the molecule. Nevertheless, we propose a tentative assignment of these bands in Appendix A.

In order to check if the samples are stable in air, we repeated the Raman measurements after 59–62 days (Appendix A). These data show that the Raman spectrum of AZPbCl_3_ shows the presence of only two new and very weak bands at 2962 and 655 cm^−1^ that could be related to an impurity phase. In the case of AZPbBr_3_, the additional bands, which appear at 2962, 1317, 677, 573, 429 and 338 cm^−1^, are much stronger. The Raman spectra of AZPbI_3_ show that the additional bands at 2944, 1372, 1302, 625, 513, 407 and 309 cm^−1^ are observed already for the freshly synthesized sample, and their intensity strongly increases with time (Appendix A). The Raman data indicate, therefore, that the air stability of the AZPbX_3_ compounds decreases in the order Cl > Br > I. As already reported by Petrosova et al., the new phases appear most likely due to the opening of the AZ^+^ ring and formation of low-dimensional perovskites based on 2-haloethylammonium cations [38].

### 2.2. Optical Properties

The RT diffuse reflectance spectra of AZPbCl_3_ and AZPbBr_3_ show the presence of sharp bands located at 415 and 553 nm, respectively (Figure 2a), which can be attributed to excitonic absorption. The obtained results were used to estimate the energy band gap (E_g_) of the investigated materials by using the Kubelka–Munk equation [47]:(1)$$F\left(R\right)=\frac{{\left(1-R\right)}^{2}}{2R}$$
where R is a reflectance. E_g_ values were determined by plotting [F(R)·hν]^2^ versus energy (hν) (Appendix A). The excitonic peaks of AZPbCl_3_ and AZPbBr_3_ were subtracted from the Kubelka–Munk function to estimate the band gap with reasonable accuracy. The diagram of the determined energy band gaps of the investigated compounds is shown in Figure 2b. As can be seen, the smallest optical band gap equal to 1.51 eV is observed for AZPbI_3_, and it increases to 2.24 eV and 3.01 eV for AZPbBr_3_ and AZPbCl_3_, respectively. The determined values are in good agreement with those reported by Petrosova et al. [38]. It is worth adding that former studies revealed that 3D FAPbX_3_ perovskites have a slightly smaller E_g_ (3.03, 2.12–2.26 and 1.36–1.51 eV for FAPbCl_3_, FAPbBr_3_ and FAPbI_3_, respectively) than MAPbX_3_ analogues (2.88–3.177, 2.15–2.392 and 1.44–1.63 eV for MAPbCl_3_, MAPbBr_3_ and MAPbI_3_, respectively) [48,49,50,51,52]. Significantly larger band gaps were, however, reported for MHy-based perovskites (3.4 and 2.58 eV for MHyPbCl_3_ and MHyPbBr_3_, respectively [8,9]). Since the values reported in the literature for the FA and MA perovskites are scattered, we measured the absorption spectra of available MA and FA analogues and compared them with those obtained for AZPbX_3_. This comparison clearly shows that both the absorption edges and excitonic bands of AZPbX_3_ compounds are significantly red shifted compared to MAPbX_3_ (Appendix A). A small red shift is also evident for AZPbBr_3_ when compared to FAPbBr_3_ (Appendix A). Thus, the energy band gap of lead halide perovskites increases in the order AZ^+^ < FA^+^ < MA^+^ < MHy^+^. The largest band gaps of MHyPbX_3_ analogues can be attributed to the large distortion of their inorganic subnetworks [8,9]. The MA, FA and AZ lead halides crystallize, however, in the ideal cubic perovskites structure at RT (except for MAPbI_3_, which, at RT, is tetragonal). Therefore, different band gaps of these compounds cannot be attributed to different degrees of octahedral distortion. Our inspection of the crystallographic data shows that MAPbX_3_ analogues have much smaller lattice parameters than the corresponding FAPbX_3_ analogues and that these parameters increase slightly when going to AZPbX_3_. Therefore, it is evident that E_g_ decreases with the increasing size of the organic cation, thus leading to an increased lattice parameter and Pb-X bond length.

The LT emission spectra of AZPbCl_3_ under 266 nm consist of an intense and narrow band at 410 nm and a broad band at 548 nm. (Figure 3a). The full width at half maximum (FWHM) of the narrow band is only 44 meV (6 nm) at 80 K. Its position overlaps with the excitonic absorption (Figure 3a). A small Stokes shift, as well as a narrow PL, is characteristic of FE recombination. Recent studies concerning MHyPbCl_3_ showed the FE band at 362 nm [9], while for MAPbCl_3_ and FAPbCl_3_, the excitonic band was located at 404 nm and near 410 nm [13,19,21]. The very large FWHM of the broad band (145 nm, 589 meV) and its large Stokes shift (133 nm) are characteristic features of STEx emission [53,54]. In the case of 3D lead halides perovskites, this type of broadband PL was reported for MHyPbCl_3_ at 512 nm [9] and for CsPbCl_3_ at 653 nm [55].

The PL spectrum of AZPbBr_3_ is red shifted compared to AZPbCl_3_ and has a band maximum at 574 nm (Figure 3). Its FWHM is 71 meV (19.9 nm). The observed band is asymmetric and consists of two narrower bands with maxima at 560 nm and 574 nm. The presence of two or even more bands was often reported for 3D lead halide perovskites, and additional bands were assigned either to the presence of trap states (bound excitons, BEs) or presence of domains with different symmetries [20,56,57,58,59,60]. For instance, studies of FAPbBr_3_ showed that, at RT, the cubic and tetragonal domains show PL at 537 and 557 nm, respectively [58], while studies of MAPbBr_3_ single crystals revealed that the spectrum at 60 K consists of a band near 544 nm attributed to FE recombination, as well as bands at 551 and 556 nm attributed to BE emission [56]. The small energy separation between the two emission bands of AZPbBr_3_ and the linear behavior of the emission intensity as a function of the laser power (Appendix A) is consistent with assignment of the higher energy band to FE recombination in the cubic phase and the lower energy band either to BE emission from the cubic phase or FE emission from a lower symmetry phase. It is worth adding that the emission of AZPbBr_3_ is red shifted compared to FAPbBr_3_, in agreement with the red shift of the excitonic absorption.

Based on the PL spectra, the CIE chromaticity coordinates of the investigated compounds were calculated and presented in Figure 3b, together with photographs of the glowing samples. These CIE coordinates correspond to yellow-green and yellow color for AZPbCl_3_ and AZPbBr_3_, respectively.

PL of AZPbI_3_ is shifted toward the infrared region. The observed band is located at 899 nm, and its FWHM is 86 meV (56.3 nm). The emission intensity of this band changes linearly with the excitation power density (Appendix A), allowing us to assign this PL to FE recombination. Similar bands were observed at 830 and 847 nm (at RT) for good-quality MAPbI_3_ and FAPbI_3_ single crystals [25,61], and near 780 and 820 nm (at 100 K) for MAPbI_3_ and FAPbI_3_ thin films, respectively, [57]. These data from the literature show that PL bands exhibit a red shift with the improving quality of the samples and when MA^+^ is replaced by FA^+^. As can be seen, the PL of AZPbI_3_ is even more red shifted than the PL of the FA analogue.

The temperature-dependent emission spectra of all aziridinium 3D lead halide perovskites show a strong influence of temperature on the position of the PL bands and their intensities (Figure 4). In the case of the chloride, the shape of the FE and STEx bands does not change upon heating, and the emission is quite stable with T_0.5_ = 150 K (Figure 4a and Figure 5). The energy activation (E_a_) for thermal quenching was estimated, using the Boltzmann relation, to be 56 meV (Appendix A). The emission of AZPbBr_3_ is less stable with the E_a_ of 29 meV and T_0.5_ of 159 K (Figure 4b and Figure 5 and Appendix A). As can be seen, a separation between the two PL bands becomes more pronounced with the increasing temperature. Furthermore, the higher energy band exhibits a significant shift toward higher energies as the temperature increases (Appendix A). It can be observed that the PL intensity of AZPbI_3_ decreases rapidly with an increasing temperature. The energy activation for thermal quenching is 40 meV, while T_0.5_ = 126 K (Figure 4c and Figure 5 and Appendix A). Similar to the bromide analogue, the FE band shifts to higher energies on heating; that is, the band maximum moves from 899 nm at 80 K to 860 nm at 300 K (Appendix A).

Studies of AZPbX_3_ samples show that, in all cases, FE band exhibits shift to higher energies upon heating and that the E_a_ values are small and fall within the 29–56 meV range. This behavior is similar to that reported for the MAPbX_3_ and FAPbX_3_ counterparts, which also showed a significant blue shift of PL upon heating [20,57,62] and small activation energies (105.21, 13–53.35, 21.67, 35 and 45 eV for MAPbCl_3_, MAPbBr_3_, FAPbBr_3_, FAPbI_3_ and MAPbI_3_, respectively [20,62,63,64].

## 3. Experimental Section

### 3.1. Synthesis of Single Crystals

Due to very high cost of aziridine and its low stability, aziridine was prepared by reacting cheap 2-bromoethylammonium hydrobromide (99%, Sigma-Aldrich, St. Louis, MO, USA) with potassium hydroxide (90%, Sigma-Aldrich), a method that was reported for the first time by Gabriel [65]. First, 15 mmol of KOH was placed in a plastic vial and dissolved in 3 mL of distilled water. Then a solution of 2-bromoethylammonium hydrobromide in 3 mL of water and 6 mL of acetonitrile (AcCN) was added. The vial was closed, shaken a few times and left at RT for 20 h (Figure 1, Solution A).

In order to synthesize AZPbBr_3_, 2 mmol of PbBr_2_ (98%, Sigma-Aldrich) was placed in a small glass vial and dissolved in 8 mL of hydrobromic acid (48%, Sigma-Aldrich). This glass vial was then placed in the plastic vial containing the abovementioned rection mixture, and the lid of the plastic vial was closed (Figure 1). Orange crystallites were separated from the solution after 2 h (Figure 1, Solution B).

AZPbI_3_ was prepared in a similar way, but the solution in a small vial contained 2 mmol of PbI_2_ (99%, Sigma-Aldrich), 2 mL of hydroiodic acid (57%, Sigma-Aldrich) and 4 mL of acetonitrile. Furthermore, the synthesis was performed at −18 °C, and the black crystallites formed on the glass vial walls were separated from the solution after 20 h.

In the case of AZPbCl_3_, the small vial contained 0.5 mmol of PbCl_2_ (98%, Sigma-Aldrich) dissolved in 4 mL of hydrochloric acid (37%, Sigma-Aldrich). Synthesis was performed at RT, and the small crystals, grown at the bottom of the glass vial, were separated after 20 h.

A good match of powder XRD patterns of AZPbBr_3_ and AZPbI_3_ with the calculated ones based on the single-crystal data reported by Petrosova et al. [38] confirmed the phase purity of the bulk AZPbBr_3_ sample and showed some minor impurity phase (peaks at 23.52, 24.82 and 30.63°) in the case of AZPbI_3_ (see Appendix A). However, it should be noticed that the peak calculated at 45.1° and observed by Petrosova et al. as a very weak peak [38] is not observed in our experimental pattern (Appendix A). The absence of this peak in our pattern is probably due to a worse signal-to-noise ratio compared to the data reported previously, thus making it difficult to observe very weak peaks. Due to the small number of AZPbCl_3_ crystals, we do not present their powder XRD pattern, but Raman spectrum confirmed the purity of this phase (Figure 1).

### 3.2. Materials and Methods

Raman spectra of powdered samples were measured by using a Bruker FT100/S spectrometer (Billerica, MA, USA) with YAG:Nd laser excitation (1064 nm). The spectral resolution was set to be 2 cm^−1^.

Powder XRD patterns were obtained by using an X’Pert PRO X-ray diffraction system (Malvern Panalytical, Malvern, UK) equipped with a PIXcel ultrafast line detector and Soller slits for CuKα_1_ radiation (λ = 1.54056 Å). The powders were measured in the reflection mode, and the X-ray tube settings were 30 mA and 40 kV.

The RT absorption spectra of the powdered samples were measured by using a Varian Cary 5E UV–Vis–NIR spectrophotometer (Varian, Palo Alto, CA, USA).

Emission spectra at various temperatures under 266 or 450 nm excitation from a laser diode were measured with the Hamamatsu photonic multichannel analyzer PMA-12, equipped with a BT-CCD linear image sensor (Hamamatsu Photonics, Iwata, Japan). The temperature of the samples was controlled by using a Linkam THMS 600 Heating/Freezing Stage (Linkam, Tadworth, UK).

## 4. Conclusions

AZPbX_3_ perovskites were synthesized by using a novel method and characterized by using Raman and optical spectroscopes. The Raman data showed that all bands above 300 cm^−1^ could be attributed to vibrations of AZ^+^ cation. Interestingly, similar to other 3D perovskites, one of the bands exhibits very strong dependence on the type of halogen ion, indicating its strong coupling with the inorganic lattice. We assigned this band to the AZ-cage mode. The Raman data also revealed that AZ^+^ cations form weaker hydrogen bonds with the halide anions than MA^+^ cations. 

The Diffuse reflectance data revealed that the excitonic absorption and PL bands, as well as band gaps of AZPbX_3_ perovskites, are red shifted compared to their MA and FA counterparts. This behavior is correlated with the size of the organic cation, which increases when going from MA^+^ to FA^+^ and then AZ^+^. The PL intensity of the studied compound strongly decreases upon heating, and the analysis of this behavior allowed us to estimate the energy activation for thermal quenching as 29, 40 and 56 meV for the Br, I and Cl analogue, respectively. These values are comparable to the exciton binding energies reported for the MA and FA lead halide perovskites. Similar to the MA and FA perovskites, the PL of AZPbX_3_ compounds also exhibited a blue shift upon heating. In conclusion, our studies show that AZPbX_3_ compounds exhibit very similar properties to those of their MA and FA counterparts. Therefore, the aziridinium analogues and AZ/MA and AZ/FA mixed-cation systems seem to be very attractive materials for light-emitting and solar cell applications.

## Data Availability

Not applicable.

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
