# Peer review of "Synthesis, Photoluminescence and Vibrational Properties of Aziridinium Lead Halide Perovskites"

_molecules, 2022, doi:10.3390/molecules27227949_

Round 1

Reviewer 1 Report

 MÄ…czka et al. systematically investigated the photoluminescence and vibrational properties (Raman) of 2-aziridinium lead halide-based 3D perovskites in this work. They found these AZPbX3 exhibit smaller band gap compared with MAPbX3 and FAPbX3 since larger cations were introduced. Overall, the manuscript is well-organized, which can be accepted after addressing the below minors:

1.     These AZPbX3 could only exist in the crystal form, which is unstable after dissolving in DMF or DMSO. Is this true? So AZPbX3 could not be solution-processed to get the film?

2.     The details of how these AZPbX3 crystals were synthesized are still not clear, and pictures or schemes need to be provided.

3.     How was the air stability of these AZPbX3 crystals?

4.     For MAPbX3 and FAPbX3, there are many phases at higher or lower temperatures, how about AZPbX3? Do they have other phases?

Author Response

Reviewer 1: : MÄ…czka et al. systematically investigated the photoluminescence and vibrational properties (Raman) of 2-aziridinium lead halide-based 3D perovskites in this work. They found these AZPbX3 exhibit smaller band gap compared with MAPbX3 and FAPbX3 since larger cations were introduced. Overall, the manuscript is well-organized, which can be accepted after addressing the below minors:

 AUTHORS: We would like to thank the reviewer for these comments.

Reviewer 1:  1.     These AZPbX3 could only exist in the crystal form, which is unstable after dissolving in DMF or DMSO. Is this true? So AZPbX3 could not be solution-processed to get the film?

AUTHORS:  We have not studied stability of these compounds in DMF or DMSO. In my opinion, aziridinium ring is opened into 2-iodoethylammonium in acidic solution (HI). The same may happen in HBr (opening into 2-bromoethylammonium) but probably to much less extent. I do not know if HCl is also able to open the ring. Anyway, all this happens in acidic solution but I think that these compound should be quite stable in neutral solutions so probably thin films can be obtained when acids are eliminated.   

Reviewer 1: 2.     The details of how these AZPbX3 crystals were synthesized are still not clear, and pictures or schemes need to be provided.

AUTHORS:  We have added a scheme showing chemical reaction, which was used for synthesis of aziridine, and method to obtain small single crystals.

Reviewer 1:  3.     How was the air stability of these AZPbX3 crystals?

AUTHORS: Stability is an important issue of perovskites. To have some information on air stability, we have measured Raman spectra of our samples, which were prepared about 2 months ago. These data were added as Figure S2 and discussion is present at the end of paragraph 3.1.

 Reviewer 1:  4.     For MAPbX3 and FAPbX3, there are many phases at higher or lower temperatures, how about AZPbX3? Do they have other phases?

AUTHORS:  Study of phase transitions is beyond scope of this paper. However, as you may noticed, AZ+ cations are disordered at RT and this fact suggests that they will order at low temperatures leading to phase transitions into ordered phases. We plan to perform temperature-dependent studies in the near future to check for presence of phase transitions.     

Reviewer 2 Report

Studies of hybrid organic-inorganic lead halide perovskites (HOIPs) attract a lot of attention due to their possible applications in optoelectronic devices, including solar cells. In the paper, the authors present experimental results relating to evaluating Raman spectra and photoluminescent properties of aziridinium lead halide perovskites. This article contains interesting results, it is well-written, and the scientific results seem sound. 

The general impression of the manuscript is good. The manuscript is properly structured and comprehensible but requires a little revision. The work is relevant. The publication adds knowledge to the field. 

I have a few points about the presentation that should be addressed prior to publication.

1.         Most of the previously performed relevant work has been cited and discussed.

Minor revision: It would be useful to include into discussion of Quarti et al. [J. Phys. Chem. Lett. 2014, 5, 279−284, dx.doi.org/10.1021/jz402589q], Ledinský et al [J. Phys. Chem. Lett. 2015, 6, 401−406, DOI: 10.1021/jz5026323], Pérez-Osorio et al [J. Phys. Chem. C 2015, 119, 25703−25718, https://doi.org/10.1021/acs.jpcc.5b07432] and Y. Kanemitsu [J. Mater. Chem. C, 2017,5, 3427-3437, https://doi.org/10.1039/C7TC00669A].

2.         Terminology is defined and used in a consistent way.

Minor revision 1: Please clarify the meaning of “AZ-cage modes” for the AZPbX3. In the case of MA-cage mode that is not only sensitive to the type of halogen anion due to coupling with the inorganic cage through N−H···X bonds, but according to Mattoni et al. [J. Phys. Chem. Lett. 2016, 7, 3, 529–535, https://doi.org/10.1021/acs.jpclett.5b02546] it is not pure torsion. The additional rationale why the authors did not attribute these modes to torsional vibration (see, e.g., Quarti et al. or Leguy et al. [Phys. Chem. Chem. Phys. 2016, 18, 27051−27066] will improve this paper and may interest a wider audience in this work.

Minor revision 2: Page 2, lines 45-46, should be free exciton (FE)  (instead of free excition (FE)).

3.         Most of the figures are valid and readable.

Minor revision: According to the paper, “The second low-wavenumber band is very sensitive to the type of halogen anion, i.e., it shifts from 468 cm-1 for AZPbCl3 to 308 cm-1 for AZPbBr3 and 240 cm-1 for AZPbI3”, but based on Figure 1 (including inset that “shows details of the low-wavenumber region”) it is impossible to distinguish a second low-wavenumber band for AZPbCl3. It would be beneficial to provide additional information on what data the statement mentioned is based on.

4.         The methodology is appropriate and properly described

Minor revision: Synthesis of single crystals - an additional explanation on whether the presence of minor peaks in the experimental XRD pattern of AZPbI3 at ~23-24° and 24-25° (Figure S1) is due to impurities will improve this paper. The absence of a peak at 45º indicates a mismatch of powder XRD patterns of AZPbI3 with the calculated and measured by Petrosova et al. [ref 38 in reviewed paper] ones. The explanation of the reasons will also improve this paper.

Author Response

Reviewer 2:  Studies of hybrid organic-inorganic lead halide perovskites (HOIPs) attract a lot of attention due to their possible applications in optoelectronic devices, including solar cells. In the paper, the authors present experimental results relating to evaluating Raman spectra and photoluminescent properties of aziridinium lead halide perovskites. This article contains interesting results, it is well-written, and the scientific results seem sound. 

The general impression of the manuscript is good. The manuscript is properly structured and comprehensible but requires a little revision. The work is relevant. The publication adds knowledge to the field. 

AUTHORS:  We would like to thank the reviser for good evaluation of our results.

Reviewer 2:  I have a few points about the presentation that should be addressed prior to publication.

  1. Most of the previously performed relevant work has been cited and discussed.

Minor revision: It would be useful to include into discussion of Quarti et al. [J. Phys. Chem. Lett. 2014, 5, 279−284, dx.doi.org/10.1021/jz402589q], Ledinský et al [J. Phys. Chem. Lett. 2015, 6, 401−406, DOI: 10.1021/jz5026323], Pérez-Osorio et al [J. Phys. Chem. C 2015, 119, 25703−25718, https://doi.org/10.1021/acs.jpcc.5b07432] and Y. Kanemitsu [J. Mater. Chem. C, 2017,5, 3427-3437, https://doi.org/10.1039/C7TC00669A]. 

AUTHORS:  We have included all suggested papers in our discussion.

Reviewer 2:  2.         Terminology is defined and used in a consistent way.

Minor revision 1: Please clarify the meaning of “AZ-cage modes” for the AZPbX3. In the case of MA-cage mode that is not only sensitive to the type of halogen anion due to coupling with the inorganic cage through N−H···X bonds, but according to Mattoni et al. [J. Phys. Chem. Lett. 2016, 7, 3, 529–535, https://doi.org/10.1021/acs.jpclett.5b02546] it is not pure torsion. The additional rationale why the authors did not attribute these modes to torsional vibration (see, e.g., Quarti et al. or Leguy et al. [Phys. Chem. Chem. Phys. 2016, 18, 27051−27066] will improve this paper and may interest a wider audience in this work.

Minor revision 2: Page 2, lines 45-46, should be free exciton (FE)  (instead of free excition (FE)).

AUTHORS:  We have clarified the meaning of AZ-cage modes, taking into account all suggested papers.

We have corrected “excition”to “exciton”.

Reviewer 2:  3.         Most of the figures are valid and readable.

Minor revision: According to the paper, “The second low-wavenumber band is very sensitive to the type of halogen anion, i.e., it shifts from 468 cm-1 for AZPbCl3 to 308 cm-1 for AZPbBr3 and 240 cm-1 for AZPbI3”, but based on Figure 1 (including inset that “shows details of the low-wavenumber region”) it is impossible to distinguish a second low-wavenumber band for AZPbCl3. It would be beneficial to provide additional information on what data the statement mentioned is based on.

AUTHORS:  We have modified Figure 1 by adding arrows indicating this highly sensitive band.

Reviewer 2:  4.         The methodology is appropriate and properly described

Minor revision: Synthesis of single crystals - an additional explanation on whether the presence of minor peaks in the experimental XRD pattern of AZPbI3 at ~23-24° and 24-25° (Figure S1) is due to impurities will improve this paper. The absence of a peak at 45º indicates a mismatch of powder XRD patterns of AZPbI3 with the calculated and measured by Petrosova et al. [ref 38 in reviewed paper] ones. The explanation of the reasons will also improve this paper.

AUTHORS:  We have added information that these weak peaks correspond to an impurity phase and that the absence of the peak near 45 º is due to its very low intensity and worse signal to noise ratio, when compared to the pattern reported by Petrosova et al.